# PHASE: Physics-Integrated, Heterogeneity-Aware Surrogates for Scientific Simulations

## Abstract

Large-scale numerical simulations underpin modern scientific discovery but remain constrained by prohibitive computational costs. AI surrogates offer acceleration, yet adoption in mission-critical settings is limited by concerns over physical plausibility, trustworthiness, and the fusion of heterogeneous data. We introduce PHASE, a modular deep-learning framework for physics-integrated, heterogeneity-aware surrogates in scientific simulations. PHASE combines data-type–aware encoders for heterogeneous inputs with multi-level physics-based constraints that promote consistency from local dynamics to global system behavior. We validate PHASE on the biogeochemical (BGC) spin-up workflow of the U.S. Department of Energy's Energy Exascale Earth System Model (E3SM) Land Model (ELM), presenting—to our knowledge—the first scientifically validated AI-accelerated solution for this task. Using only the first 20 simulation years, PHASE infers a near-equilibrium state that otherwise requires more than 1,200 years of integration, yielding an effective reduction in required integration length by at least 60×. The framework is enabled by a pipeline for fusing heterogeneous scientific data and demonstrates strong generalization to higher spatial resolutions with minimal fine-tuning. These results indicate that PHASE captures governing physical regularities rather than surface correlations, enabling practical, physically consistent acceleration of land-surface modeling and other complex scientific workflows.

## 1 Introduction

Numerical simulations, mainly grounded in domain knowledge and partial differential equations (PDEs), are fundamental pillars of modern scientific discovery, driving advances in fields from climate modeling to materials design (Hao et al., 2024; Koehler et al., 2024; Danabasoglu et al., 2020; Pathak et al., 2020; Reichstein et al., 2019). Despite their power, these simulations face a critical bottleneck: prohibitive computational cost. This burden is especially acute for tasks requiring long integration times to reach equilibrium or extensive ensemble runs for uncertainty quantification, which can consume millions of core-hours and hinder the pace of research (Bauer et al., 2015; Golaz et al., 2019; Keyes et al., 2013). Furthermore, the monolithic nature of many simulation codes makes it challenging to rapidly integrate new mechanistic processes or modify variables, slowing the cycle of scientific innovation and scientific discovery (Willard et al., 2022).

To overcome these computational barriers, AI- and ML-based surrogate models have emerged as a promising alternative (Lu & Ricciuto, 2019; Sun et al., 2023; Willard et al., 2022). By learning complex input-output mappings from simulation data, these surrogates can accelerate inference by orders of magnitude. However, their adoption in mission-critical scientific domains has been stymied by significant concerns over trustworthiness and physical plausibility (Karpatne et al., 2017; Willard et al., 2022). Purely data-driven models, which are not governed by numerical equations, can produce physically inconsistent or unrealistic results, especially when extrapolating beyond their training distribution. For instance, even state-of-the-art models like Pangu-Weather can generate non-physical artifacts, limiting their reliability for scientific inquiry (Bi et al., 2023).

In response, the community has developed Physics-Informed Neural Networks (PINNs) that embed PDE-based constraints directly into the loss function to enforce physical laws (Raissi et al., 2019; Karniadakis et al., 2021). While a conceptual advance, this approach often introduces its own

Figure 1: The PHASE surrogate accelerates the E3SM Biogeochemical (BGC) spin-up, reducing the required integration length by $60\times$ compared to the traditional simulation.

challenges. PINNs can be difficult to scale to complex, large-scale systems and can be rigid, struggling to handle the heterogeneous data types (e.g., time-series, spatial fields, layered variables) that are ubiquitous in scientific datasets (Lahat et al., 2015; Rudy et al., 2019). This creates a critical research gap: a need for a framework that unifies the efficiency of data-driven methods with the rigor of physical constraints, while also offering the flexibility to manage real-world, heterogeneous scientific data.

To address this gap, we introduce **PHASE (Physics-Integrated, Heterogeneity-Aware Surrogates for Scientific Simulations)**, a novel deep learning framework designed to build trustworthy and efficient surrogates. PHASE features a modular architecture that explicitly handles data heterogeneity through type-aware encoders, such as LSTMs for time-series data and CNNs for layered soil inputs. Crucially, it integrates domain knowledge through a multi-level hierarchy of physics-based constraints. These range from *hard constraints* implemented via architectural choices (e.g., using a Softplus activation to enforce non-negativity of physical quantities) to *soft constraints* incorporated into the loss function to penalize violations of governing physical principles. This unique combination ensures both predictive accuracy and physical consistency.

We demonstrate the power and practicality of PHASE by tackling a notoriously difficult computational bottleneck: the Biogeochemical (BGC) spin-up in the U.S. Department of Energy's Energy Exascale Earth System Model (E3SM) Land Model (ELM) (Golaz et al., 2022; 2019). This process, which requires over 1,200 years of simulation to initialize the land surface to a near-equilibrium state, is a major impediment to climate research. Using only the first 20 years of simulation data as input, PHASE accurately infers a near-equilibrium state, achieving an effective reduction in the required integration time of at least $60\times$. As conceptually illustrated in Figure 1, this represents—to our knowledge—the first validated AI-accelerated solution for this critical scientific task. The surrogate-generated states are physically plausible and can be used to successfully restart the numerical simulation. Moreover, the trained model generalizes strongly to higher spatial resolutions with minimal fine-tuning, providing compelling evidence that PHASE learns the underlying physical regularities of the system rather than superficial data correlations.

Our contributions are threefold:

- **A scalable data pipeline** that fuses complex, heterogeneous, and unaligned multi-modal ELM simulation data into a unified training dataset, addressing a core challenge in scientific AI.

- **The novel PHASE framework,** a modular architecture that integrates multi-level physics (via hard and soft constraints) with data-type–aware encoders to ensure physical plausibility while handling heterogeneous inputs.

- **First scientifically validated AI surrogate** of E3SM BGC spin-up workflow, achieving a $60\times$ reduction in simulation time and demonstrating strong generalization to higher-resolution data.

## 2 RELATED WORK

**Data-Driven Surrogates in Scientific Computing.** Machine learning surrogates have emerged as powerful tools to accelerate scientific simulations across domains such as climate, fluid dynamics, and materials design (Lu & Ricciuto, 2019; Sun et al., 2023; Willard et al., 2022; Meng et al., 2020; Yool et al., 2020). By approximating complex input–output mappings, they enable tasks like pa-

rameter estimation and ensemble prediction at much lower computational cost (Yang & Perdikaris, 2019; Rudy et al., 2019). Recent large-scale models such as Pangu-Weather (Bi et al., 2023) and FourCastNet (Pathak et al., 2022) demonstrate impressive skill in global weather forecasting by learning directly from reanalysis data. Despite their success, these purely data-driven surrogates are not governed by physical equations, and thus can generate non-physical artifacts or unstable long-term dynamics. More broadly, unconstrained learning risks spurious correlations and unreliable extrapolation beyond training distributions (Karpatne et al., 2017; Willard et al., 2022). These limitations highlight the need for frameworks that embed physical knowledge into learning processes to enhance trustworthiness in mission-critical scientific applications.

**Physics-Integrated Neural Networks.** Recent advances incorporate physical laws directly into neural models (Wu et al., 2024; Duan et al., 2025). Physics-informed neural networks (PINNs) constrain outputs to satisfy PDEs at collocation points (Raissi et al., 2019), while Region Optimized PINN (RoPINN) improves generalization by enforcing constraints on local neighborhoods (Wu et al., 2024). Other work embeds differentiable solvers as modules within networks for stability and efficiency (Chalapathi et al., 2024). Operator-learning approaches, such as the Fourier Neural Operator (FNO) (Li et al., 2021), learn mappings between function spaces with resolution-invariant properties, establishing themselves as strong surrogates for PDE-governed systems. Despite these advances, most physics-integrated models are tailored to single-task or homogeneous data, limiting their applicability to real-world scientific workflows that require multi-task predictions and heterogeneous data integration.

**Multi-Task Learning and Heterogeneous Data Fusion.** Multi-task learning (MTL) leverages shared representations to improve efficiency across related tasks (Ruder, 2017; Sun et al., 2020; Sener & Koltun, 2018; Von Rueden et al., 2021; Gao et al., 2024; Hemker et al., 2024; Ren et al., 2024), while multi-fidelity surrogate models exploit cross-resolution data to enhance accuracy (Meng et al., 2020). In scientific modeling, however, outputs span diverse structures (scalars, time series, spatial fields), and inputs include multimodal forcings, layered soil variables, and categorical plant functional type distributions. Naïve feature concatenation is often inadequate (Baltrušaitis et al., 2018), motivating more advanced approaches such as tensor fusion Hou et al. (2019), cross-attention mechanisms (Ma et al., 2023; Hemker et al., 2023), and latent-space discretization. Nevertheless, these fusion strategies rarely incorporate physics-based constraints or leverage prior domain knowledge to guide interpretable variable groupings and reduce spurious feature selection (Mosqueira-Rey et al., 2023; Geneva & Zabaras, 2019). Addressing these gaps requires unified frameworks that combine heterogeneous data fusion, multi-task prediction, and physics integration, motivating the design of PHASE.

## 3 METHODOLOGY

### 3.1 PHASE OVERVIEW

Large-scale scientific models, such as land surface models in Earth system modeling, adopt a data-centric paradigm, where water, energy, and nutrients are continuously transformed, transferred, and redistributed across diverse pools and states. These processes capture the intricate exchanges between the terrestrial surface and the atmosphere, resulting in high-dimensional, heterogeneous variable sets $\mathcal{X}$ that pose unique challenges for efficient simulation and learning. However, their significant computational demands in terms of time and resources present a major bottleneck. To mitigate these limitations, we introduce PHASE, an AI-driven trustworthy framework designed for accelerated multi-task scientific simulation. Its modular architecture is depicted in Figure 2. PHASE synergistically integrates (i) data-type-sensitive processing tailored for heterogeneous inputs, (ii) physics-based constraints $\mathcal{C}_{\text{phys}}$, and (iii) the integration of foundational scientific knowledge, denoted as $\mathcal{K}_{\text{domain}}$. Crucially, $\mathcal{K}_{\text{domain}}$ represents a set of established scientific priors and principles that are incorporated at the design stage. This knowledge is introduced as a systematic, upfront step to ensure the framework is grounded in scientific reality. This combination enables PHASE to achieve high-fidelity results in complex scientific applications. For clarity, we use Biogeochemical (BGC) models, which simulate the cycling of chemical elements through Earth's systems, as a running example to illustrate PHASE's functionality. However, PHASE's design is inherently flexible and generalizable to other computationally demanding simulators.

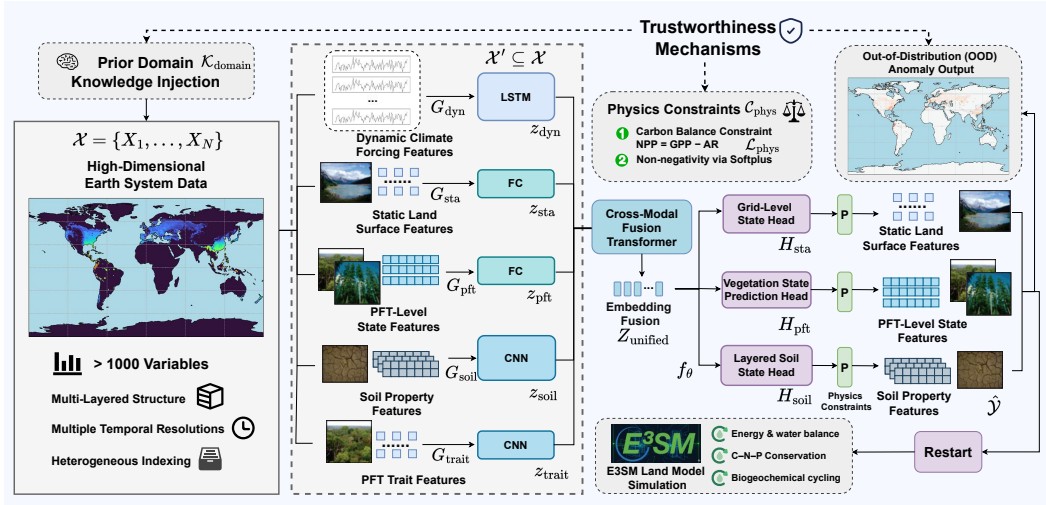

Figure 2: The physics-integrated and heterogeneity-aware architecture of PHASE.

The core objective of the PHASE framework is to learn a complex mapping function $f_\theta$ from a curated subset of heterogeneous input features, denoted as $\mathcal{X}' \subseteq \mathcal{X}$, to $M$ distinct downstream task predictions $\hat{\mathcal{Y}} = \{\hat{Y}_1, \hat{Y}_2, \ldots, \hat{Y}_M\}$. This learning process, parameterized by $\theta$, explicitly incorporates physics-based constraints $\mathcal{C}_{phys}$ and leverages prior domain knowledge $\mathcal{K}_{domain}$. This relationship is formalized as $\hat{\mathcal{Y}} = f_\theta(\mathcal{X}' \mid \mathcal{C}_{phys}, \mathcal{K}_{domain})$. Conceptually, as illustrated in Figure 2, $\mathcal{K}_{domain}$ influences feature selection and grouping at the input stage, while $\mathcal{C}_{phys}$ constrains the model's learning process and architecture through loss terms or structural rules that reflect physical laws.

## 3.2 KNOWLEDGE INTEGRATION

Large-scale scientific models often involve thousands of input variables, denoted as $\mathcal{X}$, due to their inherent scale and complexity. Using this full set without discrimination can lead to overfitting, prohibitive computational costs, and diminished interpretability. To address this challenge, our framework employs a **knowledge-guided feature engineering strategy** that leverages prior domain knowledge ($\mathcal{K}_{domain}$). This strategy curates a focused subset of variables $\mathcal{X}' \subseteq \mathcal{X}$ that are causally relevant to the simulation objectives and further organizes them into $N_g$ meaningful groups $\mathcal{G} = \{G_1, \ldots, G_{N_g}\}$ where each group $G_k \in \mathcal{G}$ comprises one or more features $X_i \in \mathcal{X}'$. Grouping is guided by physical semantics, data types, or other established scientific criteria—for example, environmental static attributes may be consolidated into one group, while time-series atmospheric forcings form another, and layered soil properties may be treated as a distinct category.

Our reproducible design integrates prior scientific principles by first defining a causality-informed feature subset ($\mathcal{X}'$). It then systematically maps data types to corresponding neural architectures, such as Long Short-Term Memory (LSTM) networks for temporal sequences and CNNs for layered spatial variables, to capture relevant correlations. This structured, knowledge-guided approach improves interpretability, and yields a transferable design that generalizes across scientific domains.

## 3.3 REPRESENTATION LEARNING

Once the knowledge-guided subset of features $\mathcal{X}'$ is organized into $N_g$ groups $\mathcal{G}$, the subsequent crucial step is to generate effective latent representations for these diverse inputs. Scientific simulation data are characterized by significant heterogeneity, encompassing varied data types (e.g., temporal sequences, spatial grids, categorical labels), structures, resolutions, and semantic interpretations within each group $G_k \in \mathcal{G}$. This complexity, which includes differing indexing schemes and spatial scales, presents challenges for creating a unified representation suitable for integrated modeling. To manage this heterogeneity, we propose a **two-stage latent representation learning process**:

### 3.3.1 MODALITY-SPECIFIC ENCODING

We employ a modular encoder architecture where each feature group $G_k$ is processed by a dedicated encoder $E_k$, tailored to its specific data structure and characteristics, as suggested by the distinct

input processing paths in Figure 2. This encoder $E_k$, parameterized by $\boldsymbol{\theta}_k$, transforms $\boldsymbol{G}_k$ into a modality-specific latent representation $\boldsymbol{z}_k$:

$$\boldsymbol{z}_k = E_k(\boldsymbol{G}_k; \boldsymbol{\theta}_k) \tag{1}$$

For example, temporal feature sequences (e.g., time-varying forcing data) are processed using Long Short-Term Memory (LSTM) networks to capture temporal dependencies; Spatially structured or multi-layer variables (e.g., soil properties with depth) are handled by Convolutional Neural Networks (CNNs) to exploit spatial or vertical correlations; Scalar or vector features without explicit sequence or spatial structure (e.g., static environmental attributes, PFT features) are embedded using Fully Connected (FC) layers. The resulting $\boldsymbol{z}_k$ is a compact, learned representation of the input feature group $\boldsymbol{G}_k$.

### 3.3.2 UNIFIED LATENT SPACE FUSION

The set of individual latent representations, $\{\boldsymbol{z}_1, \boldsymbol{z}_2, \ldots, \boldsymbol{z}_{N_g}\}$, often resides in different embedding spaces and thus cannot be directly combined. Naive fusion strategies, such as simple concatenation or averaging, fail to capture complex cross-modal interactions and typically underutilize the complementary information encoded in each modality. To overcome this, we introduce a dedicated fusion module $F_{\text{fusion}}$, parameterized by $\boldsymbol{\phi}_{\text{fusion}}$, which builds upon a Transformer encoder to dynamically integrate heterogeneous features into a shared latent manifold.

The Transformer-based fusion mechanism leverages multi-head self-attention to learn pairwise dependencies across modalities, enabling each embedding $\boldsymbol{z}_k$ to attend to others in a context-dependent manner. This allows the model to emphasize informative relationships (e.g., between climate forcings and vegetation traits) while suppressing spurious correlations. Positional and modality-specific encodings are incorporated to preserve the structural identity of each group $\boldsymbol{G}_k$, ensuring that temporal features, static attributes, and layered soil states are distinguished within the fusion process. By jointly modeling interactions across all groups, the fusion module produces a contextually enriched latent representation:

$$\boldsymbol{Z}_{\text{unified}} = F_{\text{fusion}}(\boldsymbol{z}_1, \boldsymbol{z}_2, \ldots, \boldsymbol{z}_{N_g}; \boldsymbol{\phi}_{\text{fusion}}) \tag{2}$$

This unified representation $\boldsymbol{Z}_{\text{unified}}$ captures higher-order correlations across heterogeneous inputs, enabling structurally diverse features to be meaningfully combined. Moreover, the modular design of $F_{\text{fusion}}$ provides flexibility: additional modalities or variable groups can be seamlessly incorporated without redesigning the overall architecture. Such adaptability is critical for scientific simulations, where new variables or higher-resolution data are often introduced, and it ensures that the fused representation remains both scalable and physically interpretable for downstream multi-task prediction and applications.

### 3.4 PREDICTION AND TRUSTWORTHINESS

A key challenge in developing AI surrogates for numerical simulations is handling the diverse nature of target outputs $\mathcal{Y}$. These outputs often vary significantly in dimensionality and structure: some are scalar values representing grid-level aggregates, others are structured vectors (e.g., depth-resolved carbon pools), and some are even matrices spanning multiple dimensions like PFTs and soil layers (e.g., soil organic matter across PFT $\times$ depth).

To effectively predict these heterogeneous targets within a single, unified model, PHASE employs a multi-task learning (MTL) framework. As depicted in Figure 2, after the unified latent representation $\boldsymbol{Z}_{\text{unified}}$ is generated, it is fed into $M$ distinct task-specific prediction heads (e.g., Grid-Level State Head, Vegetation State Prediction Head, Layered Soil State Head). Each head, $H_j$ (parameterized by $\boldsymbol{\psi}_j$), is tailored to predict a specific target output $\hat{\boldsymbol{Y}}_j$:

$$\hat{\boldsymbol{Y}}_j = H_j(\boldsymbol{Z}_{\text{unified}}; \boldsymbol{\psi}_j) \quad \text{for } j = 1, \ldots, M \tag{3}$$

Figure 2 illustrates this with a multi-task perceptron having dedicated heads. For instance, a scalar branch might predict low-dimensional continuous variables, a vector branch could produce structured 1D outputs (e.g., vertical profiles in BGC), and a matrix branch might generate 2D outputs. Each branch typically uses dedicated fully connected layers (within the Multi-Task Perceptron block) to project $\boldsymbol{Z}_{\text{unified}}$ into the appropriate target shape, potentially followed by reshaping operations to restore the physical layout. This MTL architecture enables the joint modeling of diverse outputs while respecting their structural constraints and enhancing physical interpretability.

The PHASE framework is trained by minimizing a composite loss function $\mathcal{L}_{\text{total}}$. This loss integrates the losses from individual prediction tasks and a physics-informed regularization term under foundational domain knowledge $\mathcal{K}_{\text{domain}}$:

$$\mathcal{L}_{\text{total}} = \sum_{j=1}^{M} w_j \mathcal{L}_{\text{task}}^{(j)} + \lambda \mathcal{L}_{\text{phys}} \tag{4}$$

where $\mathcal{L}_{\text{task}}^{(j)}$ is the loss for the $j$-th task, $w_j$ is its corresponding weight (e.g., $w_j = 1$ for all tasks if equally weighted), and $\lambda$ is a hyperparameter balancing the contribution of the physics-based constraint loss $\mathcal{L}_{\text{phys}}$.

Each task-specific output $\hat{\boldsymbol{Y}}_j$ is typically supervised using a regression loss, such as the Mean Squared Error (MSE), comparing the prediction with the ELM simulation results $\boldsymbol{Y}_j$:

$$\mathcal{L}_{\text{task}}^{(j)}(\hat{\boldsymbol{Y}}_j, \boldsymbol{Y}_j) = \frac{1}{N_s^{(j)}} \sum_{s=1}^{N_s^{(j)}} \left\| \hat{\boldsymbol{y}}_s^{(j)} - \boldsymbol{y}_s^{(j)} \right\|_2^2 \tag{5}$$

Here, $\hat{\boldsymbol{y}}_s^{(j)}$ and $\boldsymbol{y}_s^{(j)}$ are the predicted and ELM simulation results for the $s$-th sample of the $j$-th task, and $N_s^{(j)}$ is the number of samples for that task.

To further instill domain consistency, we incorporate $\mathcal{L}_{\text{phys}}$, a physics-informed soft constraint. Using our BGC example, a fundamental equation governing the plant carbon balance is that Net Primary Productivity (NPP), which is the net carbon assimilated by plants, is equal to Gross Primary Productivity (GPP), the total carbon captured through photosynthesis, minus Autotrophic Respiration (AR), the carbon lost as plants respire. This is expressed as $NPP = GPP - AR$. This relationship is enforced as a soft constraint by directly penalizing deviations:

$$\mathcal{L}_{\text{phys}} = \frac{1}{N_{\text{samples}}} \sum_{i=1}^{N_{\text{samples}}} \left( \hat{y}_{\text{NPP}}^{(i)} - \left( \hat{y}_{\text{GPP}}^{(i)} - \hat{y}_{\text{AR}}^{(i)} \right) \right)^2 \tag{6}$$

where $\hat{y}_{\text{NPP}}^{(i)}$, $\hat{y}_{\text{GPP}}^{(i)}$, and $\hat{y}_{\text{AR}}^{(i)}$ are the model's predictions for these specific quantities for the $i$-th sample, and $N_{\text{samples}}$ is the total number of samples over which this constraint is applied.

Our framework ensures the trustworthiness and physical plausibility of its predictions through three core, integrated mechanisms: (1) **Prior Domain Knowledge Injection**, where the model architecture is fundamentally grounded in scientific principles ($\mathcal{K}_{\text{domain}}$) by mapping causally relevant variables to specialized encoders that respect the physical nature of the data; (2) **Physics-Informed Constraints**, where the learning process is guided by explicit physical laws ($\mathcal{C}_{\text{phys}}$) through both *soft constraints*, such as adding a penalty term to the loss function for the carbon balance equation ($NPP = GPP - AR$), and *hard constraints*, such as using a Softplus activation function to enforce non-negativity; and (3) **Automated Anomaly Detection**, an Out-of-Distribution (OOD) mechanism flags predictions in uncertain regions as a safety check.

## 4 EXPERIMENTAL RESULTS

### 4.1 DATASET AND EVALUATION

We constructed a large-scale, unified training dataset from complex global ELM simulations, covering 20,975 land grid cells at a $1°$ resolution (Golaz et al., 2022). The primary challenge was fusing heterogeneous data from multiple sources (e.g., history, restart, and forcing files), which we addressed using a custom data pipeline to create a cohesive, grid-cell-centered dataset. Leveraging domain prior knowledge, input features were categorized into five major groups to align with PHASE's modular architecture: (1) Dynamic Climate Forcing Features, (2) Static Land Surface Features, (3) Plant Functional Type (PFT) Trait Features, (4) PFT-Level State Features, and (5) Layered Soil and Dead Organic Matter Features. Full details on the data pipeline and features are available in Appendix A.

To evaluate the model's performance in accelerating the Biogeochemical (BGC) spin-up, we selected six key variables that are highly representative of an ecosystem's slow-turnover equilibrium state:

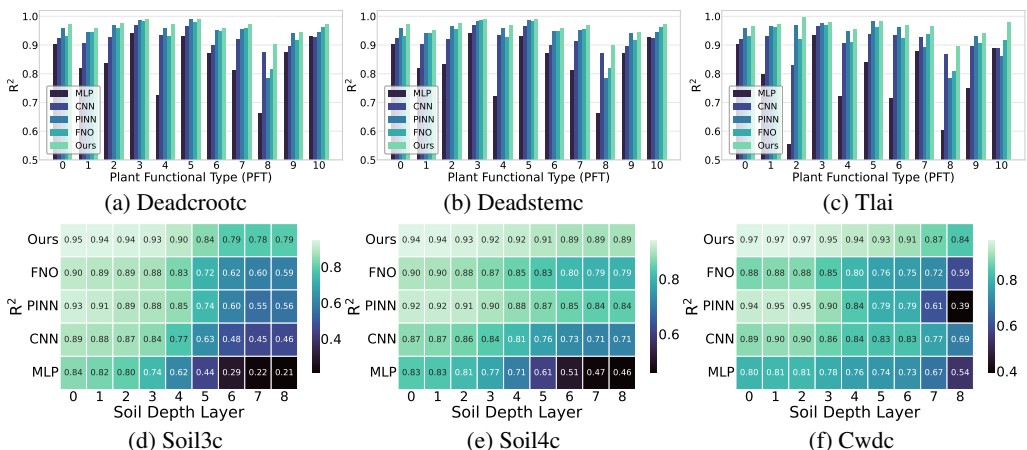

Figure 3: $R^2$ ($\uparrow$) performance on PFT- and soil depth-structured outputs.

three dead organic matter pools, `Deadcrootc` (Dead Coarse Root Carbon), `Deadstemc` (Dead Stem Carbon), and `Cwdc` (Coarse Woody Debris Carbon); two sequentially-linked soil carbon pools, `Soil3c` and `Soil4c`; and a key vegetation state indicator, `Tlai` (Total Leaf Area Index).

The selection of these specific variables is central to our acceleration strategy. These pools are the slowest-moving components in the land model, and their stabilization dictates the multi-centennial timescale of the BGC spin-up. Our model, PHASE, is designed to infer the near steady-state values for only these slow processes. Crucially, PHASE does not generate a complete restart file. Instead, a subsequent, shorter ELM run is necessary to allow faster-moving variables to equilibrate with the AI-inferred states. We integrate these AI-augmented outputs into the restart file and then perform a 100-year simulation to achieve a fully stable and consistent steady state. This two-stage approach provides a comprehensive assessment of the final equilibrium while reducing the required simulation time by at least $60\times$.

## 4.2 COMPARISON WITH BASELINE MODELS

To evaluate PHASE, we benchmark against representative baselines: Multilayer Perceptron (MLP), Convolutional Neural Network (CNN), physics-informed (PINN), and operator-learning (FNO) models. As the first surrogate tailored for BGC spin-up, these baselines were adapted for our multi-task prediction scenario (Appendix B).

Table 1 illustrates that PHASE achieves the highest coefficient of determination $R^2$ scores across nearly all variables and uniquely provides stable restart capability. While PINN and FNO deliver competitive accuracy, they often generate physically implausible values (e.g., negative soil carbon pools) that cause ELM simulations to crash. In contrast, PHASE's modular, data-type-sensitive architecture ensures physically valid outputs, enabling successful restart files. The corresponding RMSE results (Appendix C) and layer-wise $R^2$ analysis (Figure 3) further confirm these advantages.

Table 1: Model performance ($R^2$ $\uparrow$) and restart capability at $1°$ resolution

| Model | Deadcrootc | Deadstemc | Tlai | Soil3c | Soil4c | Cwdc | Restart[a] |
|---|---|---|---|---|---|---|---|
| MLP | $0.7865_{\pm0.0450}$ | $0.7852_{\pm0.0480}$ | $0.7500_{\pm0.0620}$ | $0.5544_{\pm0.0550}$ | $0.6650_{\pm0.0610}$ | $0.7370_{\pm0.0850}$ | $\times$ |
| CNN | $0.9230_{\pm0.0310}$ | $0.9206_{\pm0.0330}$ | $0.9098_{\pm0.0380}$ | $0.6962_{\pm0.0410}$ | $0.7951_{\pm0.0450}$ | $0.8322_{\pm0.0780}$ | $\times$ |
| FNO | $0.9412_{\pm0.0028}$ | $0.9413_{\pm0.0030}$ | $0.9222_{\pm0.0010}$ | $0.7714_{\pm0.0041}$ | $0.8475_{\pm0.0018}$ | $0.8002_{\pm0.0134}$ | $\times$ |
| PINN | $0.9445_{\pm0.0035}$ | $0.9432_{\pm0.0031}$ | $0.9359_{\pm0.0041}$ | $0.7680_{\pm0.0290}$ | $0.8813_{\pm0.0117}$ | $0.7955_{\pm0.1089}$ | $\times$ |
| Ours | $\mathbf{0.9649_{\pm0.0036}}$ | $\mathbf{0.9637_{\pm0.0040}}$ | $\mathbf{0.9651_{\pm0.0043}}$ | $\mathbf{0.8733_{\pm0.0012}}$ | $\mathbf{0.9146_{\pm0.0034}}$ | $\mathbf{0.9274_{\pm0.0022}}$ | $\checkmark$ |

[a] Capability to generate a stable state for restarting simulations.

Figure 4 illustrates the spatial and PFT-dimensional agreement between predictions and ELM simulations, showing only minor and spatially sparse deviations, together with strong consistency across PFTs. These results demonstrate that PHASE preserves both spatial patterns and PFT-dependent distributions of soil carbon with high fidelity. See Appendix D for results on other key variables.

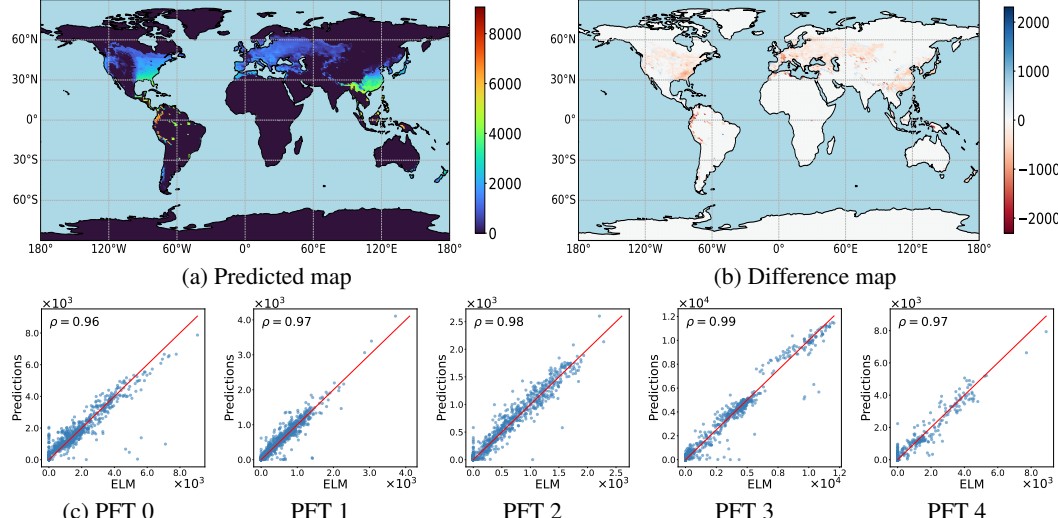

(a) Predicted map        (b) Difference map

(c) PFT 0     PFT 1     PFT 2     PFT 3     PFT 4

Figure 4: Analysis of `Deadcrootc` predictions. (a-b) Spatial comparison for the representative PFT 0. (c) Scatter plots across the first five Plant Functional Types (PFTs 0-4).

## 4.3 QUANTIFYING THE IMPACT OF DOMAIN KNOWLEDGE

To assess the role of domain knowledge, we conducted an internal comparison between two versions of PHASE. The first version (**Base**) was trained without incorporating phosphorus-related information when constructing the dataset. The second version (**Base + P**) explicitly included prior knowledge that phosphorus is a limiting nutrient in many ecosystems. The results strongly support our hypothesis that integrating such domain knowledge is critical for model accuracy. As shown in the latitudinal error distributions (Figure 5), the Base model exhibits its largest discrepancies in the tropical and subtropical zones. In contrast, Base + P demonstrates a markedly narrower error distribution and reduced bias in these regions. This visual evidence confirms that including scientifically critical variables is essential for building a robust surrogate model that avoids regional biases. For a detailed quantitative analysis, the layer-by-layer soil carbon predictions are provided in Appendix E (Table 6). Beyond integrating domain knowledge, we further evaluate whether PHASE 's outputs can be seamlessly embedded into real ELM workflows.

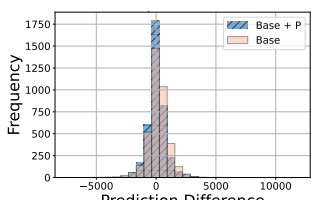

Figure 5: Soil3c error distribution in Tropics ($< 23°$).

## 4.4 WORKFLOW FOR ACCELERATED BGC SPIN-UP

Our approach yields substantial reductions in computational cost compared to conventional numerical simulations. A standard 1200-year spin-up at 1-degree resolution typically requires around 40 hours on a high-performance computing system using 10 nodes and 1,280 CPU cores. In contrast, our optimized PHASE framework completes training in approximately 1 hour and inference in under 10 minutes on a single NVIDIA A100 GPU. The workflow proceeds as follows. First, a 20-year simulation provides the input data for PHASE, which infers near steady-state values for slow processes. It is important to note that PHASE focuses only on these slow-turnover variables and is not designed to create a complete restart file. Therefore, a subsequent ELM run is necessary to allow the faster-moving variables to equilibrate with the AI-inferred values. We integrate these AI-augmented outputs into the restart file and then perform a 100-year simulation to achieve a fully stable and consistent steady state. This approach reduces the initial long-duration simulation time by at least $60\times$. This capability to produce functionally valid and numerically stable initial conditions, rather than merely generating offline predictions, is a critical differentiator of our approach. It validates that PHASE captures the underlying physical regularities of the system, enabling its direct integration into mission-critical scientific workflows. This confirms the trustworthiness of its predictions, a feat not achieved by the baseline models evaluated in this study.

## 4.5 GENERALIZATION ACROSS SPATIAL RESOLUTIONS

To evaluate the generalization capability of our framework, we examine its performance when transferring from a $1°$ training resolution to a $0.5°$ dataset, constructed following the same procedure outlined in Section 4.1. Three settings are compared: Zero-shot, directly applying the $1°$ model; Few-shot, fine-tuning with 5% or 10% of $0.5°$ samples; and Full-train, conventional training with an 80/20 split. Table 2 shows averaged $R^2$ scores (with RMSE in Appendix C, Table 5). Zero-shot yields much lower accuracy, while Few-shot rapidly recovers performance, approaching Full-train results. This demonstrates that although cross-resolution transfer is challenging, limited fine-resolution data suffice to adapt the model by adjusting scale-specific details. PHASE thus captures resolution-invariant ecological and physical patterns, enabling efficient adaptation to higher resolutions with minimal data.

Table 2: Generalization performance ($R^2 \uparrow$) at $0.5°$ resolution

| Method | Deadcrootc | Deadstemc | Tlai | Soil3c | Soil4c | Cwdc |
|---|---|---|---|---|---|---|
| Zero-shot | $0.7293_{\pm 0.0009}$ | $0.7098_{\pm 0.0016}$ | $0.5139_{\pm 0.0002}$ | $0.3428_{\pm 0.0001}$ | $0.4001_{\pm 0.0000}$ | $0.5973_{\pm 0.0003}$ |
| Few-shot (5%) | $0.9056_{\pm 0.0000}$ | $0.9018_{\pm 0.0000}$ | $0.8402_{\pm 0.0000}$ | $0.8225_{\pm 0.0000}$ | $0.8825_{\pm 0.0000}$ | $0.8482_{\pm 0.0000}$ |
| Few-shot (10%) | $0.9310_{\pm 0.0036}$ | $0.9291_{\pm 0.0039}$ | $\mathbf{0.8527_{\pm 0.0595}}$ | $0.8346_{\pm 0.0346}$ | $0.8798_{\pm 0.0388}$ | $0.8479_{\pm 0.0172}$ |
| Full-train | $\mathbf{0.9571_{\pm 0.0093}}$ | $\mathbf{0.9594_{\pm 0.0063}}$ | $0.8467_{\pm 0.0030}$ | $\mathbf{0.8716_{\pm 0.0119}}$ | $\mathbf{0.9265_{\pm 0.0049}}$ | $\mathbf{0.8910_{\pm 0.0048}}$ |

## 4.6 ABLATION STUDIES

We performed ablation studies to assess the contribution of each model component by systematically removing the CNN branch, the fully connected (FC) branch, the LSTM module, the Transformer encoder, and the physics-based loss term $\mathcal{L}_{\mathrm{phys}}$. As shown in Table 3, removing the CNN branch causes the largest degradation, especially for structured outputs such as `Tlai` and long-term soil carbon pools, underscoring the need for detailed state initialization. The FC branch contributes moderately, while the LSTM proves essential for capturing temporal dynamics across most targets. Excluding the Transformer notably reduces accuracy for Cwdc and soil pools, confirming the importance of cross-branch feature fusion. Finally, removing $\mathcal{L}_{\mathrm{phys}}$ results in small but consistent declines, highlighting the stabilizing effect of physics-based regularization.

Table 3: Ablation study of PHASE components ($R^2 \uparrow$) when removing the CNN, FC, LSTM, Transformer encoder (Trans.), or physics-based loss $\mathcal{L}_{\mathrm{phys}}$.

| Variable | Ours | w/o CNN | w/o FC | w/o LSTM | w/o Trans. | w/o $\mathcal{L}_{\mathbf{phys}}$ |
|---|---|---|---|---|---|---|
| Deadcrootc | **0.964** | $0.719_{-0.246}$ | $0.960_{-0.005}$ | $0.960_{-0.004}$ | $0.948_{-0.016}$ | $0.961_{-0.003}$ |
| Deadstemc | **0.963** | $0.717_{-0.246}$ | $0.957_{-0.007}$ | $0.960_{-0.004}$ | $0.947_{-0.016}$ | $0.959_{-0.004}$ |
| Tlai | **0.965** | $0.689_{-0.276}$ | $0.949_{-0.016}$ | $0.955_{-0.009}$ | $0.961_{-0.004}$ | $0.961_{-0.004}$ |
| Soil3c | **0.873** | $0.824_{-0.049}$ | $0.867_{-0.006}$ | $0.839_{-0.034}$ | $0.853_{-0.020}$ | $0.868_{-0.005}$ |
| Soil4c | **0.914** | $0.868_{-0.046}$ | $0.919_{+0.005}$ | $0.904_{-0.010}$ | $0.899_{-0.015}$ | $0.906_{-0.008}$ |
| Cwdc | **0.927** | $0.910_{-0.017}$ | $0.912_{-0.014}$ | $0.906_{-0.021}$ | $0.848_{-0.079}$ | $0.921_{-0.006}$ |

## 5 CONCLUSION

We introduced PHASE, a physics-integrated, heterogeneity-aware framework for building trustworthy AI surrogates of complex scientific simulations. By combining a unified data construction pipeline, data-type–sensitive encoders, and multi-level physics constraints, PHASE achieves both predictive accuracy and physical plausibility. Applied to the computationally intensive Biogeochemical spin-up in ELM, it is the first scientifically validated AI surrogate to reduce integration length by over 60× while producing stable restart states and generalizing effectively to higher spatial resolutions. These results demonstrate that PHASE captures underlying physical regularities rather than surface correlations, providing a practical and trustworthy acceleration tool for Earth system modeling. Future work will focus on enhancing scalability, extending to broader science applications, and further strengthening physical consistency and interpretability in AI-driven scientific emulation.

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

## A  DATA PREPARATION AND IMPLEMENTATION DETAILS

### A.1  DATASET CONSTRUCTION PIPELINE

The construction of the training dataset involved a multi-stage pipeline designed to unify and align data from various ELM simulation outputs:

- **Multi-Source Integration and Indexing:** The primary difficulty was fusing data from various files (e.g., history files, restart files, surface data, and forcing data). We first identified all valid land points. For PFT-based and column-based variables not indexed by grid cell, we created an **inverted mapping** to link each grid cell ID to its list of PFT or soil column indices, preserving the full vertical structure for layered variables.

- **Spatial and Temporal Alignment:** Since forcing data grids do not perfectly align with the ELM model grid, we used an efficient KD-Tree nearest-neighbor search to map each land grid cell to its closest forcing grid index. Raw 6-hourly time series data was aggregated into monthly averages to align with other variables, while static surface properties were read directly using their grid-cell indices.

- **Batch Processing and Finalization:** To manage the high dimensionality, the processed samples were ordered by latitude and longitude and saved into spatially coherent batches. Finally, these multi-modal outputs were standardized into tensors with consistent shapes for efficient training. A similar dataset was constructed at a $0.5°$ resolution using the same methodology to evaluate generalization capabilities.

### A.2  FEATURE SELECTION AND PREPROCESSING

To ensure data quality and effective model training, several preprocessing steps were applied:

- **Data Cleaning:** Non-physical data points were meticulously removed; this included variables associated with invalid PFTs and carbon pool values reported from implausibly deep soil layers.

- **Normalization:** All input features and target labels underwent MinMax normalization. This standardization enforces physically realistic value bounds and improves model convergence during training.

- **Data Partitioning:** The final curated dataset was randomly shuffled and partitioned into training and testing sets using an 80:20 ratio, resulting in 16,780 samples for training and 4,195 for testing.

### A.3 Feature Grouping

Leveraging domain prior knowledge and in alignment with PHASE's modular architecture, the input features were categorized into five major groups. This categorization ensures that variables of similar physical meaning and data structure are consistently encoded by their respective neural network modules. The groups are described below:

- **Dynamic Climate Forcing Features:** This group includes time-series meteorological drivers such as radiation, precipitation, pressure, humidity, and near-surface air temperature. These variables represent the external forcings that regulate energy and water exchange between the atmosphere and land surface.
- **Static Land Surface Features:** These features describe slowly varying or invariant characteristics of each grid cell, including geographical location, land fraction, soil texture, soil phosphorus pools, and vegetation cover fractions. They define the environmental context in which dynamic processes occur.
- **Plant Functional Type (PFT) Trait Features:** This group captures biophysical and biochemical traits of different PFTs, such as C:N ratios, photosynthetic pathway (C3/C4), leaf and root turnover, and canopy reflectance parameters. These features are essential for representing vegetation heterogeneity across ecosystems.
- **PFT-Level State Features:** These variables characterize vegetation states that evolve during the simulation, such as total leaf area index (`Tlai`), dead stem carbon (`Deadstemc`), and dead coarse root carbon (`Deadcrootc`). They provide direct indicators of vegetation structure and turnover.
- **Layered Soil and Dead Organic Matter Features:** This group contains vertically structured pools such as coarse woody debris carbon (`Cwdc`), soil carbon pool 3 (`Soil3c`), and soil carbon pool 4 (`Soil4c`). These represent the slowest-turnover components of the terrestrial carbon cycle and are critical for determining long-term ecosystem equilibrium.

## B  Baseline Model Implementations

### B.1  Fourier Neural Operator (FNO) Baseline

Specifically, for the FNO baseline, we adopted a unified operator learning paradigm where independent Fourier Neural Operator modules were constructed for each heterogeneous input type (time-series, static, 1D, and 2D variables). For non-grid data such as static attributes, we converted them into one-dimensional sequences via a broadcasting strategy to fit the FNO framework. Features extracted from each branch were then fused through a Multi-Layer Perceptron (MLP) before being passed to the multi-task prediction heads. We chose FNO as a baseline as it represents a state-of-the-art approach for learning resolution-invariant operators for physical systems, making it a strong and relevant benchmark for our task.

### B.2  Physics-Informed Neural Network (PINN) Baseline

For the PINN baseline, we implemented a physics-informed method based on state evolution. The model was designed to not only predict the final equilibrium state but also concurrently predict the physical change ($\Delta$-state) from the initial state. Its composite loss function included both a data loss, which supervises the accuracy of the final state prediction, and a physics loss, which supervises the accuracy of the predicted state change. This dual objective guides the model to learn solutions consistent with the intrinsic evolution principle: $State_{\text{final}} = State_{\text{initial}} + \Delta State$. This specific PINN variant was chosen to directly test the effectiveness of supervising the system's temporal dynamics, providing a clear comparison against the constraint-based physics integration within our proposed PHASE model.

## C  Additional Experimental Results

This section provides supplementary results that complement the main findings presented in the paper. Specifically, we report the Root Mean Square Error (RMSE) metrics, which offer an alternative

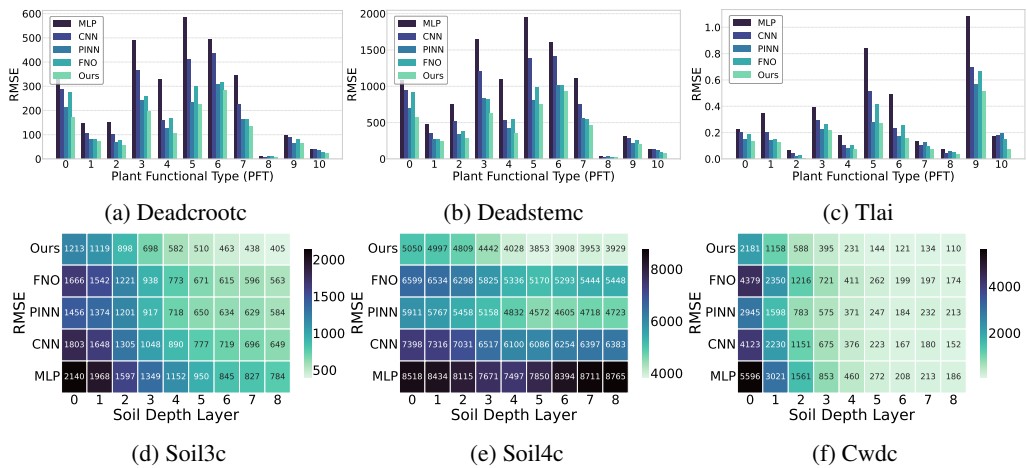

Figure 6: RMSE ($\downarrow$) performance on PFT- and soil depth-structured outputs at $1°$ resolution.

perspective on model performance to the $R^2$ scores discussed in the main text. **Table 4** details the layer-averaged RMSE for each model at the $1°$ resolution, and **Figure 6** visualizes these results, breaking them down by Plant Functional Type (PFT) and soil depth. These findings corroborate the conclusions from the $R^2$ analysis, underscoring the superior performance of our proposed model. Furthermore, **Table 5** presents the RMSE results for the generalization experiments at the $0.5°$ resolution, aligning with the $R^2$ data shown in the main body and further demonstrating the model's robustness across different spatial scales.

Table 4: Model performance (RMSE $\downarrow$) at $1°$ resolution

| Model | Deadcrootc | Deadstemc | Tlai | Soil3c | Soil4c | Cwdc |
|---|---|---|---|---|---|---|
| MLP | $274.296_{\pm 27.40}$ | $926.387_{\pm 92.60}$ | $0.6516_{\pm 0.0652}$ | $1290.270_{\pm 116.10}$ | $8217.080_{\pm 739.50}$ | $1374.266_{\pm 123.70}$ |
| CNN | $202.413_{\pm 18.20}$ | $683.529_{\pm 61.50}$ | $0.3629_{\pm 0.0327}$ | $1059.566_{\pm 95.40}$ | $6609.147_{\pm 594.80}$ | $1030.718_{\pm 92.80}$ |
| FNO | $161.257_{\pm 2.18}$ | $538.615_{\pm 9.51}$ | $\mathbf{0.2121_{\pm 0.0024}}$ | $959.439_{\pm 7.77}$ | $5772.886_{\pm 1.42}$ | $1135.439_{\pm 48.75}$ |
| PINN | $139.965_{\pm 2.37}$ | $480.38_{\pm 10.52}$ | $0.2544_{\pm 0.0063}$ | $906.876_{\pm 40.60}$ | $5082.625_{\pm 210.96}$ | $794.161_{\pm 375.27}$ |
| Ours | $\mathbf{121.065_{\pm 2.14}}$ | $\mathbf{410.173_{\pm 8.69}}$ | $0.2300_{\pm 0.0035}$ | $\mathbf{702.885_{\pm 12.80}}$ | $\mathbf{4329.755_{\pm 99.44}}$ | $\mathbf{562.641_{\pm 7.53}}$ |

Table 5: Generalization performance (RMSE $\downarrow$) at $0.5°$ resolution

| Method | Deadcrootc | Deadstemc | Tlai | Soil3c | Soil4c | Cwdc |
|---|---|---|---|---|---|---|
| Zero-shot | $383.63_{\pm 0.10}$ | $1391.78_{\pm 0.70}$ | $2358.70_{\pm 1.28}$ | $1578.84_{\pm 0.36}$ | $10513.94_{\pm 1.61}$ | $0.6182_{\pm 0.0005}$ |
| Few-shot 0.05 | $184.90_{\pm 0.00}$ | $626.45_{\pm 0.00}$ | $1021.41_{\pm 0.00}$ | $744.85_{\pm 0.00}$ | $4608.20_{\pm 0.00}$ | $0.2862_{\pm 0.00}$ |
| Few-shot 0.1 | $125.92_{\pm 1.29}$ | $419.74_{\pm 1.73}$ | $\mathbf{777.96_{\pm 56.90}}$ | $621.44_{\pm 31.90}$ | $4025.00_{\pm 26.05}$ | $0.2019_{\pm 0.0713}$ |
| Full-train | $\mathbf{101.51_{\pm 14.65}}$ | $\mathbf{338.48_{\pm 58.09}}$ | $956.12_{\pm 29.27}$ | $\mathbf{572.30_{\pm 5.14}}$ | $3555.38_{\pm 90.68}$ | $\mathbf{0.1963_{\pm 0.0091}}$ |

## D  MODEL PERFORMANCE VISUALIZATION

This section provides a detailed visual assessment of the PHASE model's performance, complementing the quantitative metrics presented earlier. **Figure 7** offers a granular inspection of model accuracy via scatter plots. The subsequent figures (**Figures 8–13**) provide a spatial evaluation for key ecosystem variables, each displaying the global predicted map alongside a difference map (prediction minus ELM simulation result). **Figures 8–10** show results for two representative Plant Functional Types (PFT 0 and PFT 4) to demonstrate performance across different vegetation communities. **Figures 11–13** show results at two representative soil depths: the biochemically active surface (Layer 0) and the mid-soil (Layer 4). These visualizations collectively demonstrate the model's high fidelity in reproducing essential geographical and structural patterns.

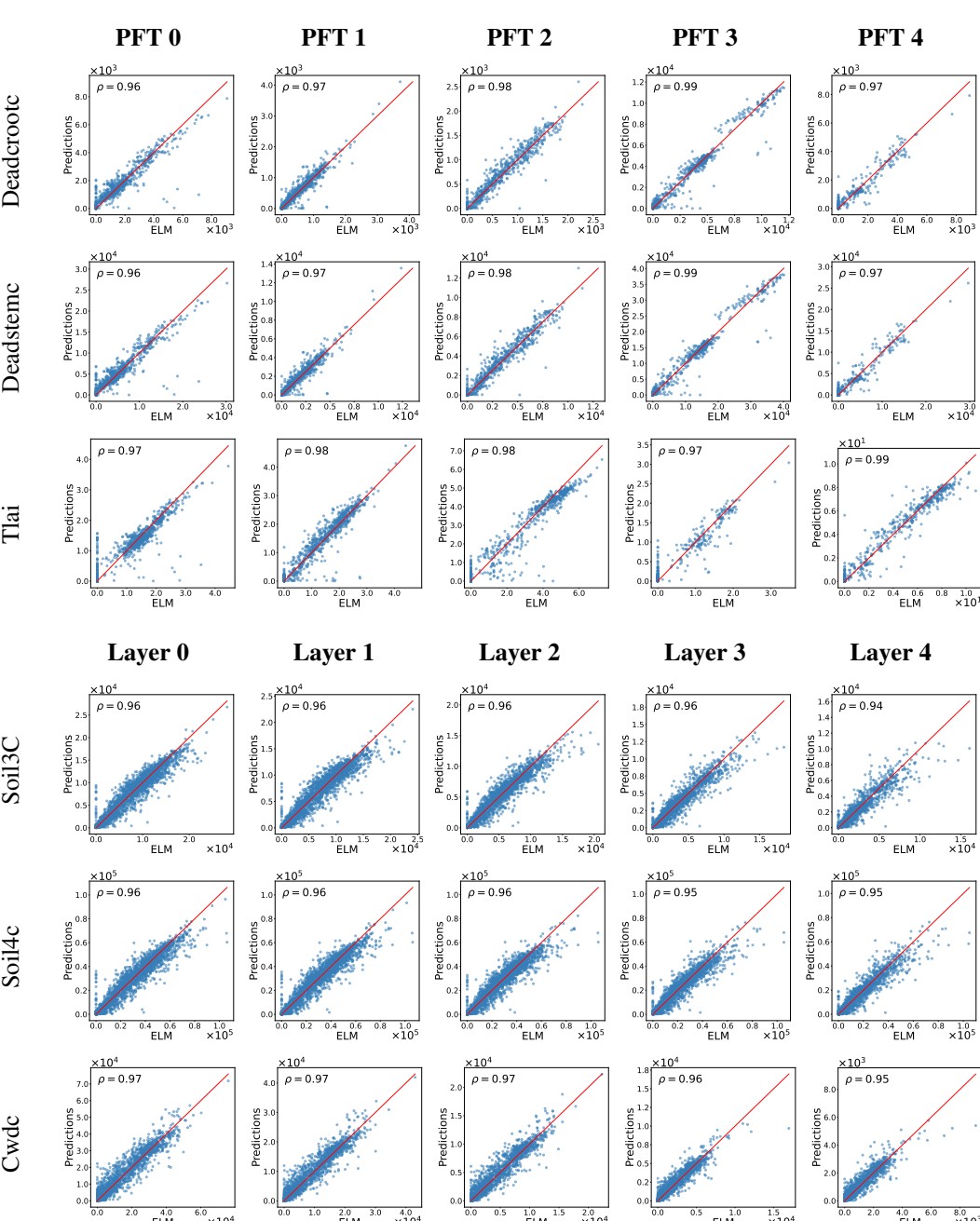

Figure 7: Scatter plot of six different variables in the top five dimensions. The first three variables are indexed by Plant Functional Type (PFT), while the last three are indexed by soil layer.

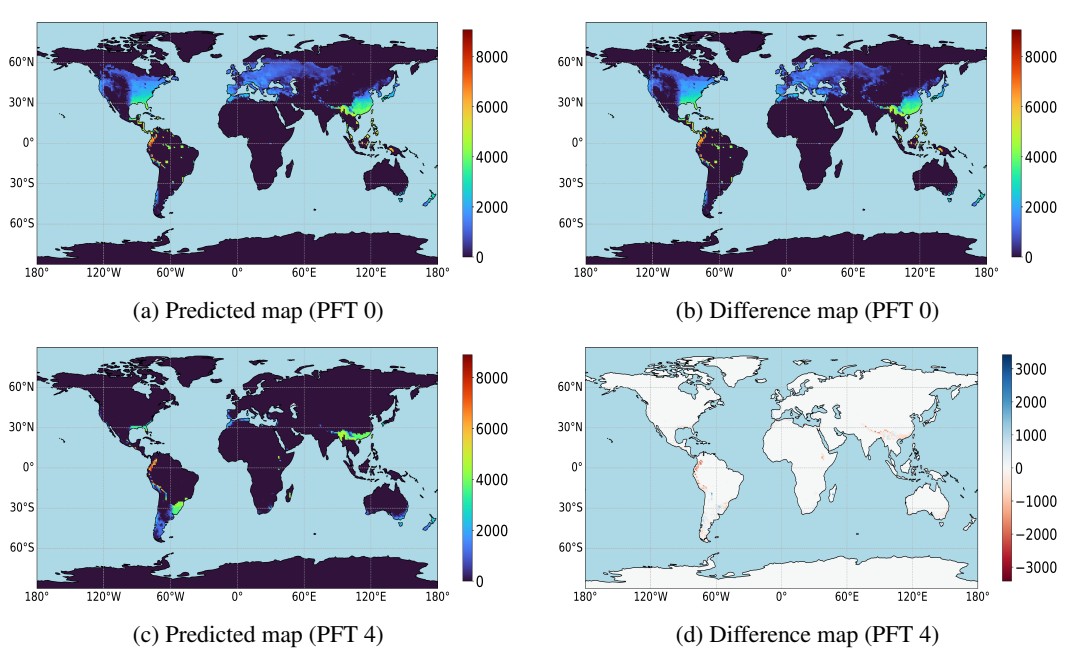

Figure 8: Spatial evaluation of `Deadcrootc` predictions for representative Plant Functional Types (PFT 0 and PFT 4).

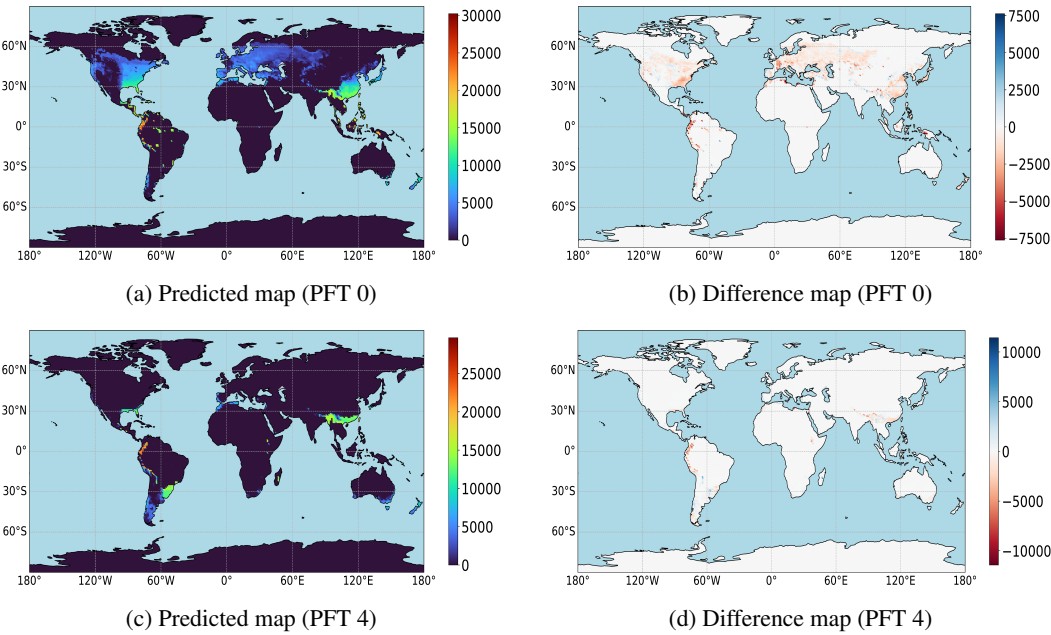

Figure 9: Spatial evaluation of `Deadstemc` predictions for representative Plant Functional Types (PFT 0 and PFT 4).

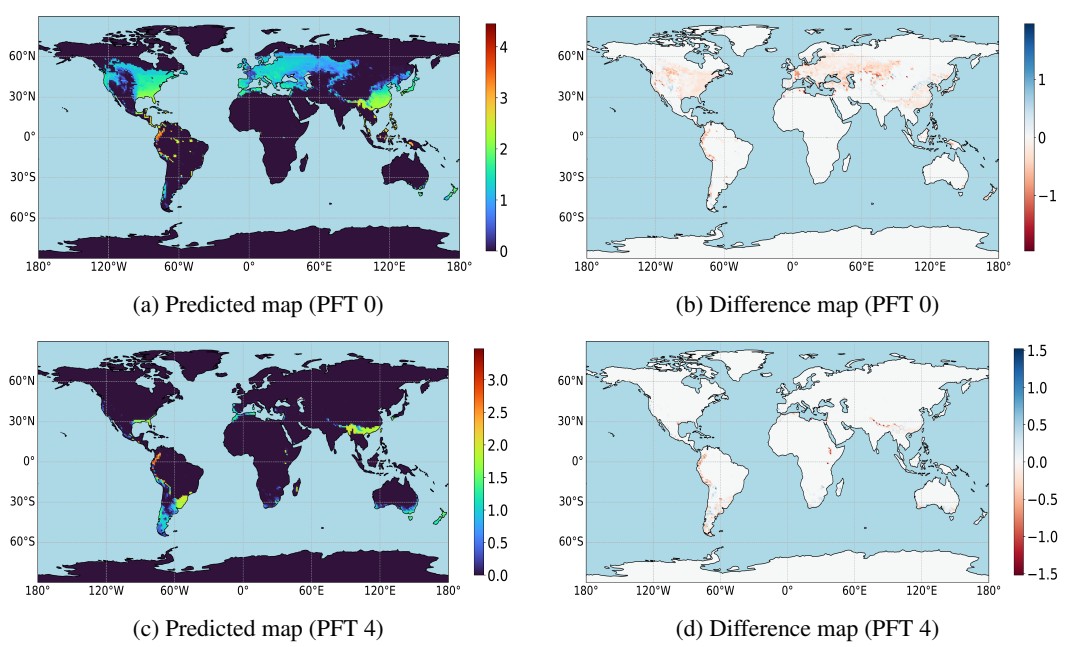

Figure 10: Spatial evaluation of `Tlai` predictions for representative Plant Functional Types (PFT 0 and PFT 4).

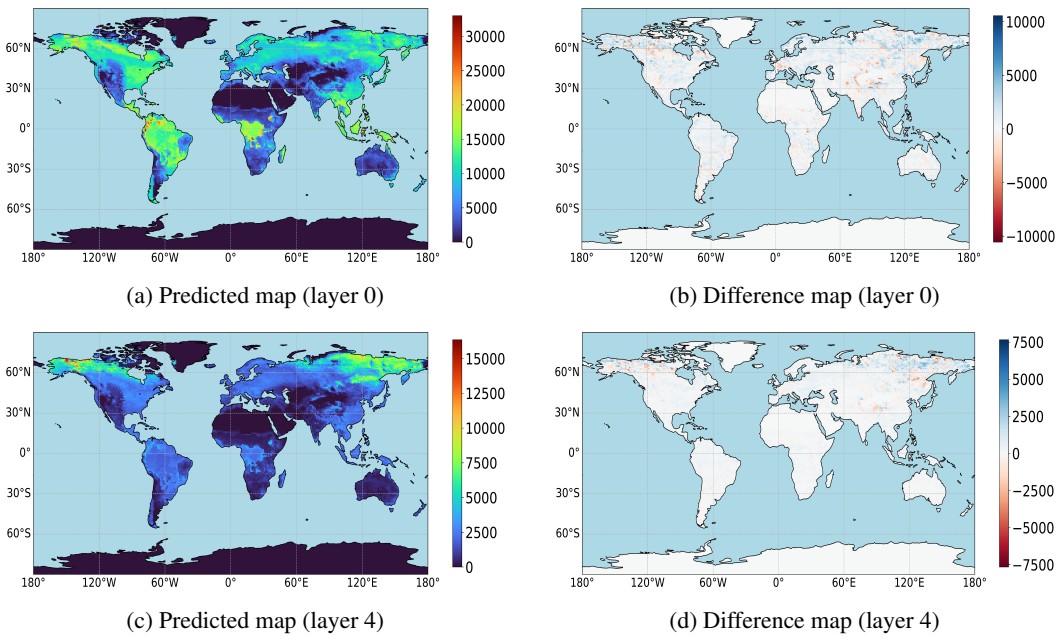

Figure 11: Spatial comparison of predicted `Soil3c` maps and difference maps for the surface layer (top) and a mid-soil layer (bottom).

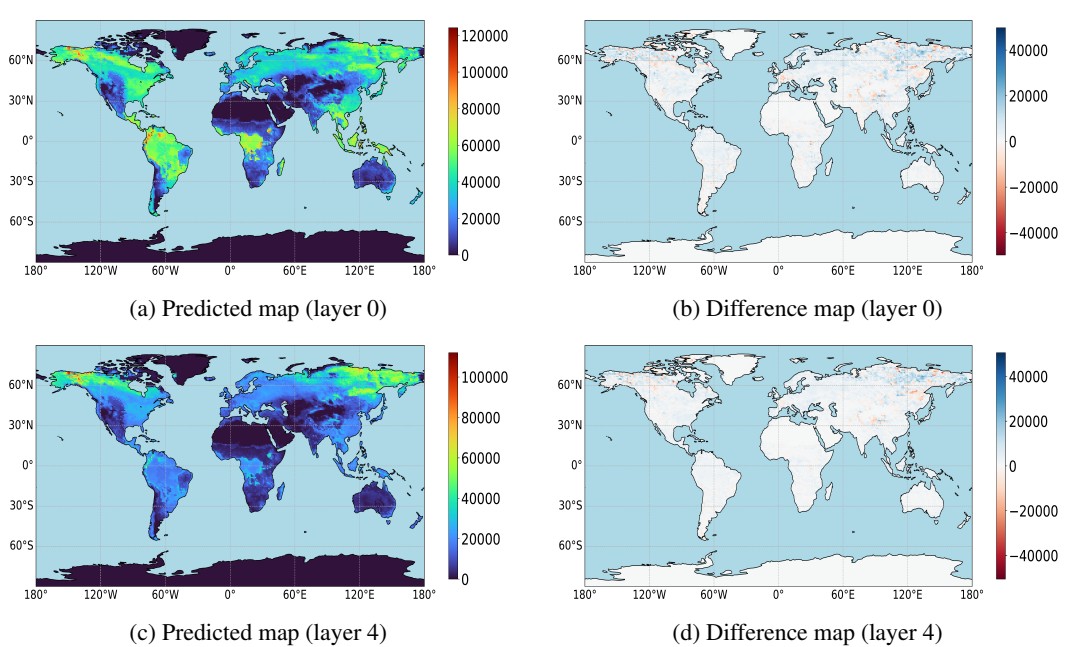

(a) Predicted map (layer 0)

(b) Difference map (layer 0)

(c) Predicted map (layer 4)

(d) Difference map (layer 4)

Figure 12: Spatial comparison of predicted `Soil4c` maps and difference maps for the surface layer (top) and a mid-soil layer (bottom).

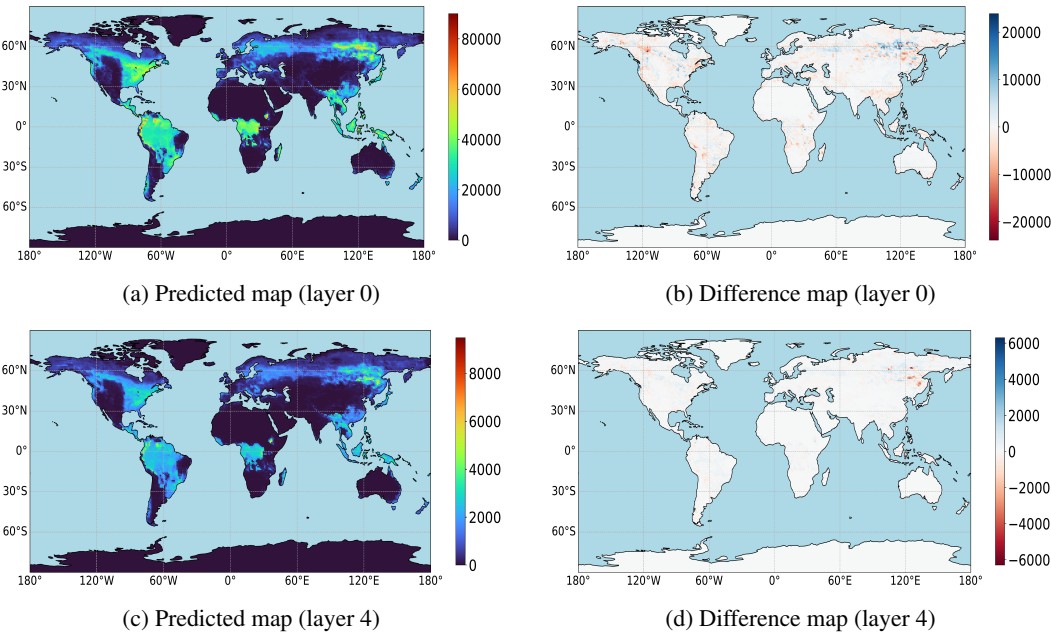

(a) Predicted map (layer 0)

(b) Difference map (layer 0)

(c) Predicted map (layer 4)

(d) Difference map (layer 4)

Figure 13: Spatial comparison of predicted `Cwdc` maps and difference maps for the surface layer (top) and a mid-soil layer (bottom).

# E    SUPPLEMENTARY DATA FOR IMPACT ANALYSIS

This section provides the detailed quantitative data supporting the analysis of scientifically-informed features presented in Section 4.3. We selected `Soil3c` and `Soil4c` for this analysis because they are the primary long-term soil carbon reservoirs, and their response directly reflects the impact of phosphorus availability on the ecosystem's carbon cycle. Table 6 presents the layer-by-layer soil carbon predictions, showing a closer alignment with the ELM simulation for the Base + P configuration.

Table 6: Impact of phosphorus data on layer-wise soil carbon prediction

| Layer | Soil3c | | | Soil4c | | |
|---|---|---|---|---|---|---|
| | **Base** | **Base + P** | **ELM** | **Base** | **Base + P** | **ELM** |
| Layer 0 | 1.11E+08 | **1.01E+08** | 9.84E+07 | 4.17E+08 | **3.92E+08** | 3.76E+08 |
| Layer 1 | 9.83E+07 | **9.06E+07** | 8.70E+07 | 4.07E+08 | **3.83E+08** | 3.66E+08 |
| Layer 2 | 7.18E+07 | **6.66E+07** | 6.33E+07 | 3.74E+08 | **3.52E+08** | 3.35E+08 |
| Layer 3 | 4.66E+07 | **4.31E+07** | 4.09E+07 | 3.13E+08 | **2.95E+08** | 2.78E+08 |
| Layer 4 | 2.89E+07 | **2.67E+07** | 2.59E+07 | 2.39E+08 | **2.24E+08** | 2.11E+08 |
| Layer 5 | 1.80E+07 | **1.65E+07** | 1.53E+07 | 1.76E+08 | **1.62E+08** | 1.54E+08 |
| Layer 6 | 1.20E+07 | **1.07E+07** | 9.78E+06 | 1.37E+08 | **1.23E+08** | 1.18E+08 |
| Layer 7 | 9.07E+06 | **7.93E+06** | 7.39E+06 | 1.17E+08 | **1.04E+08** | 9.99E+07 |
| Layer 8 | 7.01E+06 | **6.33E+06** | 5.79E+06 | 1.08E+08 | **9.57E+07** | 9.08E+07 |
| **Sum**[*] | 4.03E+08 | **3.70E+08** | 3.54E+08 | 2.28E+09 | **2.13E+09** | 2.02E+09 |

[*]*The 'Sum' row represents the total soil carbon stock across all vertical layers.*

