# OpenReview forum: "PHASE: Physics‑Integrated, Heterogeneity‑Aware Surrogates for Scientific Simulations"
_ICLR.cc/2026/Conference — ICLR 2026 Conference Withdrawn Submission_

### Official Review · Reviewer_FbK7 · 2025-10-20

**Soundness:** 2
**Presentation:** 2
**Contribution:** 2
**Rating:** 2
**Confidence:** 3

**Summary:**

This paper introduces PHASE, a deep learning framework for building physically consistent AI surrogates of large-scale scientific simulations. The approach integrates domain knowledge and multi-level physical constraints (both hard and soft) with data-type–specific encoders (CNNs, LSTMs, FC layers) to handle heterogeneous scientific data. PHASE is demonstrated on the biogeochemical spin-up problem in the DOE E3SM Land Model (ELM), showing that it can infer a near-equilibrium state using only 20 simulation years instead of 1,200, achieving a 60× acceleration while maintaining physical plausibility.

**Strengths:**

- The work tackles a well-known computational bottleneck in Earth system modeling (BGC spin-up), providing a tangible example of scientific acceleration with potential for broad application in other simulation-heavy fields.

- The modular design combining modality-aware encoders, Transformer-based fusion, and hierarchical physics constraints is technically sound and addresses key limitations of PINNs and operator-learning models.

- The simulation speed improvement is timely but is expected with ML emulators.

**Weaknesses:**

- While well-engineered, PHASE mostly repackages existing ideas (physics-informed loss, multimodal encoders, MTL) into one framework. Conceptually, it reads more as an engineering synthesis than a fundamentally new methodology.

- All experiments focus on a single case (E3SM BGC spin-up). It’s unclear how the framework would generalize to other domains or PDE-based systems beyond land surface modeling.

- Despite claiming “physics integration,” the paper offers minimal formalism or analysis of how the physics constraints improve learning dynamics or guarantee consistency. The constraint examples (e.g., NPP = GPP – AR) are simplistic and ad hoc.

- Also the claim of converging to equilibrium is not well justified, is it possible that ML model, trained on MSE, will tend to the mean value much faster anyway? I'm not sure how tending to equilibrium is interesting if not tested against e.g., tipping point or pacemaker experiments (+2K, +4K temperature forcing and their impacts on Amazon dieback)? This experiment will show if the equilibrium state reached is even accurate (due to learning the right physics), and not just merely a feature of capturing the training dataset climatology.

**Questions:**

See above section on weakness.

---

### Official Review · Reviewer_ky3X · 2025-10-25

**Soundness:** 2
**Presentation:** 3
**Contribution:** 2
**Rating:** 4
**Confidence:** 4

**Summary:**

This paper proposes a modular surrogate for E3SM-ELM BGC spin-up: heterogeneous encoders (LSTM/CNN/FC), transformer fusion, physics soft/hard constraints (e.g., Softplus for non-negativity; NPP = GPP − AR penalty). Claims >60× acceleration by inferring slow pools from 20 years of data, then restarting ELM for 100 years to reach equilibrium. Outperforms MLP/CNN/FNO/PINN on R²/RMSE, generalizes from 1° → 0.5° with few-shot fine-tuning, and can produce restartable states (baselines often crash).

**Strengths:**

Practical, high-impact workflow contribution: generating restartable states that actually run forward in ELM is a meaningful bar beyond offline metrics.

Tackles heterogeneous inputs/targets systematically; the architecture diagram and data pipeline are helpful.

Solid ablations (CNN/LSTM/Transformer/physics-loss) and cross-resolution study.

The domain-knowledge example (adding P in the tropics) is a nice, concrete illustration.

**Weaknesses:**

Physics integration is relatively light-touch.
The current mechanism is primarily soft constraints + architectural non-negativity. There’s no explicit conservation enforcement, differentiable constraints, or coupling to a solver. For a paper emphasizing “physics-integrated,” this is closer to regularized MTL than to, say, constrained operator learning. Please position claims accordingly and/or add stronger physics integration (e.g., constrained decoding, projected gradients, or differentiable diagnostic modules).

Restart validation needs hard numbers.
The headline is the 60× speedup via restartability. Please report quantitative drift metrics after the 100-year continuation: e.g., tendencies of target pools, global/biome-wise mass balance errors, and failure rate across sites. Right now, the restart result is demonstrated but not statistically characterized.

Uncertainty & trust.
The use-case (initializing a coupled Earth system model) begs for uncertainty estimates (epistemic/aleatoric) and calibration checks. The OOD “anomaly” mechanism is mentioned, but thresholds, scoring, and deployment policy aren’t described. Even simple ensembles or MC-dropout with reliability diagrams would help.

Data access & reproducibility.
The dataset is built from internal ELM runs; without a releasable subset or scripts, end-to-end replication is hard. Consider releasing (i) the fusion/encoder templates, (ii) a toy public analog (e.g., open land model components), and (iii) restart evaluation scripts.

Comparators & fairness.
FNO and a state-evolution PINN are relevant, but this task (restartable steady states) is unusual. Could you include a hybrid surrogate with constrained outputs (e.g., learned decoder followed by a small balance-enforcing reconciliation), or a solver-in-the-loop variant as a stronger baseline?

**Questions:**

Are the physics constraints hard-enforced during decoding, or only penalized via loss? What is the magnitude of constraint violation at inference time?
How is the trade-off parameter between physics and data loss tuned? Is it static or adaptive (e.g., gradient-balancing)?
How does the anomaly detector quantify distribution shift? What metric (e.g., Mahalanobis, reconstruction error) is used, and how is the threshold calibrated?
How transferable is PHASE to other PDE-based simulators (e.g., ocean, atmosphere)? Would the same modular hierarchy apply?

---

### Official Review · Reviewer_krjj · 2025-11-01

**Soundness:** 2
**Presentation:** 2
**Contribution:** 1
**Rating:** 2
**Confidence:** 4

**Summary:**

The paper proposed PHASE, a deep learning method to emulate the E3SM Land Model (ELM) for the biogeochemical (BGC) spin-up workflow. The proposed method employs various encoder and prediction head architectures to handle multi-modal data and imposes soft constraints on the prediction. PHASE performs better than simple baselines such as CNN, FCN, and PINN.

**Strengths:**

- The paper is original to the best of my knowledge.
- The paper is easy to read.

**Weaknesses:**

Overall, I think the paper proposes a well-engineered model that works relatively well for a specific problem and dataset, but does not present any significant contribution to the field of machine learning in general or scientific simulations in particular.
- The knowledge integration in Section 3.2 is rather manually crafted, data-dependent, and not generally applicable to other datasets or tasks. This seems to be going in opposite directions from what's known to work well with DL: end-to-end learning systems.
- Using different architectures for different types of input data (2nd para of Section 3.2) is a relatively trivial and standard practice when dealing with different data types. I do not consider this a contribution.
- Using different prediction heads for different types of output is also a standard practice.
- The physics-informed soft constraint is also data- and task-dependent and does not apply to other simulation problems.
- The baselines are weak and outdated. How does PHASE compare with more modern architectures used for weather and climate simulation [1, 2, 3, 4, 5]?

Other comments:
- The paper misses important global weather forecasting papers in the related work section [1, 2, 3, 4].
- The encoding-and-fusion approach was previously proposed in ClimaX [5] and Stormer [4] for weather forecasting.

[1] Lam, Remi, et al. "Learning skillful medium-range global weather forecasting." Science 382.6677 (2023): 1416-1421.

[2] Chen, Lei, et al. "FuXi: a cascade machine learning forecasting system for 15-day global weather forecast." npj climate and atmospheric science 6.1 (2023): 190.

[3] Kochkov, Dmitrii, et al. "Neural general circulation models for weather and climate." Nature 632.8027 (2024): 1060-1066.

[4] Nguyen, Tung, et al. "Scaling transformer neural networks for skillful and reliable medium-range weather forecasting." Advances in Neural Information Processing Systems 37 (2024): 68740-68771.

[5] Nguyen, Tung, et al. "ClimaX: A foundation model for weather and climate." International Conference on Machine Learning. PMLR, 2023.

**Questions:**

- How does one apply the proposed method to other datasets or simulation tasks?

---

### Official Review · Reviewer_LqjA · 2025-11-01

**Soundness:** 3
**Presentation:** 2
**Contribution:** 2
**Rating:** 6
**Confidence:** 4

**Summary:**

The authors propose PHASE, a framework for learning fast physical simulators that can handle different types of input data, incorporate physical constraints and priors, and provide more realistic simulations than purely data-driven approaches. PHASE groups its input based on prior knowledge and feeds it into a suitable network architecture to obtain latent variables. These latent variables are then fused using a transformer encoder. Based on this unified latent space, different heads predict different output variables.
The authors test their approach using a complex BGC simulation, demonstrating its efficacy.

**Strengths:**

A general approach with strong speed up, which could help other areas in science to speed up their research as well.

**Weaknesses:**

* Figure 2 is too cluttered and contains too many details, which makes it much harder to understand. Simplifying the figure and adding more description could be a solution.
* Not particularly new: handling multi-model data has already been extensively studied in the context of large language models, for example. Incorporating domain knowledge and constraints via architecture and losses has also already been done.
* Only one major simulator was addressed. Other simulations would also be interesting, as it was stated that PHASE is a general approach.

**Questions:**

* What is the exact architecture of the model used in the paper, e.g. the number of neurons per layer, the number of channels and the activation functions? Have you tested different hyperparameters? What is the trade-off between model efficiency and accuracy when these parameters are varied?
* The MLP and CNN baselines are conceptually very simple. Currently, transformers are considered state of the art. How would PHASE perform against transformers?

---

### Note · Authors · 2025-11-23

I have read and agree with the venue's withdrawal policy on behalf of myself and my co-authors.